

# Acclimatizing Fast Orthogonal Search (FOS) Model for River Stream-flow Forecasting

Abdalla Osman[1], Mohammed Falah Allawi[2], Haitham Abdulmohsin Afan[2], Aboelmagd Noureldin[1], Ahmed El-shafie[3]

[1] Electrical and Computer Engineering, Royal Military College of Canada, Kingston, Canada
[2] Civil and Structural Engineering Dept. Universiti Kebangsaan Malaysia, Malaysia
[3] Department of Civil Engineering, Faculty of Engineering, University Malaya

*Correspondence to*: Haitham Abdulmohsin Afan (haitham.afan@gmail.com)

**Abstract.** River stream-flow is well-thought-out as an essential element in the hydrology studies,
especially for reservoir management. Forecasting river stream-flow is the key for the hydrologists in proposing certain short or long-term planning and management for water resources system. In fact, developing stream-flow forecasting models are generally categorized into two main classes; process and data-driven model.

Different model techniques based on empirical methods, such as stochastic model or regression model,
more recently, Artificial Intelligent (AI) models have been examined and could provide accurate stream-flow forecasting. However, AI models experienced crucial difficulty is the necessity to utilize appropriate pre-processing methods for the raw data. In addition, the AI model should be augmented with proper optimization model to adjust the model parameters to achieve the optimal accuracy.

In this paper, a novel model namely; Fast Orthogonal Search (FOS) model is proposed to develop river
stream-flow forecasting. FOS is basically structured for recognizing the difference equation and its functional expression model for the mapping between the model input and output. The major advantage of using FOS is the waiver of the requirement of data pre-processing and optimization model for model parameters adjustment as these procedures are performed implicitly inside FOS. In addition, pole-zero cancellation procedure within FOS process can detect the over-fitted models and avoid them.



The proposed FOS method was adopted in this research to perform stream-flow forecasting model at Aswan High Dam using monthly basis for130 years. Results showed outstanding performance for stream-flow forecasting accuracy compared to other AI models developed during the last 10 years.

**Keywords:** Fast Orthogonal Search; stream-flow forecasting; Aswan High Dam; River Nile; Multi-Lead forecasting

# 1    Introduction

## 1.1    Background

In order to have optimal water resources management and planning, it is vital to have an accurate forecasting model for river stream-flow. With the purpose of having optimal planning and management of water resources system, it is vital to have an accurate forecasting model for river stream-flow. In addition, stream-flow forecasting helps in understanding and estimate suspended sediment, generation of the hydropower, design proper irrigation system for particular crop pattern and optimal release policy from reservoirs (Kisi and Cimen, 2011; Liu et al., 2014; Terzi and Ergin, 2014; Wu et al., 2009, 2005).in addition, having long-term accurate stream-flow forecasting provides the decision-makers of water resources with a valuable machination to reduce the side effect of flooding on infrastructures and human life (Box and Jenkins, 1970; Kagoda et al., 2010; Kalman and others, 1960; Kisi, 2010; Salas et al., 1980; Shiri and Kisi, 2010; Valipour, 2015; Valipour et al., 2012).

Models of river stream-flow for forecasting purposes are generally classified into two categories: process-driven models and data-driven models (Billings, 2013; Veiga et al., 2014). Process-driven models combine empirical and physically based equations to represent the physical controlling processes of the system mathematically. Data-driven models are only based on empirical equations calibrated to field data (Billings, 2013; Veiga et al., 2014). For real-time forecasting, data-driven models might be favorable, as complex physical models regularly required great amounts of information data and time-consuming computational for model simulation.



The problem of river flow forecasting can be generally formulated in nonlinear system identification problem (Chen and Dyke, 2007). The basic delinquent in the utilization of artificial intelligent methods, such as ANFIS and ANN, is the non-standardization set of rules on optimizing the implementation of them. In case that the model is straightforward, the results might be outlying to generalized in other cases;

whereas in case the model is very dense there may be deficient in generalizing, and its architecture parameters could be more challenging to simulate and elucidate. In fact, unpretentious models comprising fewer parameters and simple mathematical procedures (e.g., ordinary least squares solution) usually might be more appropriate for stream-flow forecasting model (Billings, 2013).

## 1.2  Problem Statement

In the last three decades, several stream-flow forecasting models have been developed utilizing several methods. Firstly, conventional methods (CMs) such as Auto Regression (AG), Moving Average (MA) and Auto Regression Moving Average (ARMA) have been examined for several case studies and show potential for accurate forecasting for medium stream-flow classes (Billings, 2013; Chen and Dyke, 2007; Clark et al., 2008; Husain, 1985; Kalman and others, 1960; Moradkhani et al., 2005; Noureldin et al.,

2007; Schreider et al., 2001; Valipour et al., 2012; Veiga et al., 2014). However, it has been reported that there are several drawbacks in developing these models (Clark et al., 2008; El-Shafie et al., 2012; Husain, 1985; Ju et al., 2009; Maier et al., 2004; Noureldin et al., 2007, 2011; Schreider et al., 2001). The main meagreness that associated to the application of CMs methods for developing the forecasting model for stream-flow is the stipulation to integrate it with a pre-formulation of the trustful stochastic model to

ascertain the source of uncertainty for the model input and output. Moreover, erstwhile analysis for the raw data related to the covariance values between all the variables used in developing the model.

In the last two decades, Artificial Intelligent (AI) models have been believed to have the potential solutions in order to overcome the drawbacks of the CMs methods. These methods have the ability to mimic the complex time series with its non-linearity features which are the major feature of any historical

stream flow records (Asadi et al., 2013; Balestrassi et al., 2009; Graves and Pedrycz, 2009; Han and Qiao,



2013; Ju et al., 2009). Additionally, during the development of the forecasting model for stream-flow utilizing AI methods, there is no need for having pre-formulation and analysis for the stochastic pattern for the raw stream-flow data.to be modeled which is usually required when to utilize the CMs methods (Danandeh Mehr et al., 2013; El-Shafie et al., 2009; Graves and Pedrycz, 2009; Guo et al., 2011; Jothiprakash and Magar, 2012; Katambara and Ndiritu, 2010; Kisi et al., 2012; Sang, 2013; Whigham and Crapper, 2001; Yaseen et al., 2015). Despite this, the AI model encountered numerous disputes in its model architecture due to the need to inaugurate it with proper pre-treatment method for data noisy reduction. In addition, an applicable optimal procedure should be augmented to accommodate the method's parameters to enhance the forecasting accuracy (Labat, 2005; Maier, Holger R and Dandy, 2000; Nourani et al., 2014; Sang, 2013).

In the last five years, the authors introduced several AI models for forecasting the stream-flow at the Aswan High Dam (AHD). These models methods include Adaptive Neuro-Fuzzy Inference System (ANFIS), Radial Basis Function Neural Network (RBFNN) and Multi-Layer Perceptron Neural Network (MLPNN) with ensemble procedure (El-Shafie and Noureldin, 2011). Actually, these models revealed appropriate prospective to achieve moderately high accuracy for stream-flow forecasting especially for most of the peak events at AHD. Noticeably, these AI models successfully provided worthwhile tendency for mimicking the input-output pattern in circumstances that no need for inclusion of any other influence parameters that affecting the value of the stream-flow. Even though these AI models demonstrate to be proficient, the convergence of the model during the training (calibration) experienced a slow procedure which means that the model falls in the sub-optimal search procedure. In addition, learning procedure algorithm of AI model for certain data pattern could experience an over-fitting problem during the training session and showed that the AI model is unable to generalize the accuracy for all pattern.

## 1.3 Motivation

In the light of the above, the limitations of the CMs and AI-based methods for developing stream-flow forecasting model motivate to search for better forecasting methods for stream-flow. Keeping in mind



that the design of the model assemblage the advantages of previous methods and overcome their drawbacks. FOS is employed in this research as a robust algorithm for building nonlinear system model (Korenberg, 1988) based on nonlinear autoregressive moving average model (NARMAX) (Billings, 2013). The NARMAX model is formed of three main terms which are the input, output and noise together

with their delayed terms (Billings, 2013). Usually, fewer terms form the model and hence the selection of model terms one at a time is more convenient for efficient operation of the modeling process. This selection technique is achieved in this research using FOS to enable the efficient building of NARMAX model for river stream flow.

Orthogonal search (Korenberg, 1988, 1989; Tseng and Powers, 1993) is a highly efficient technique

designed originally for efficient nonlinear system modeling. The modeling process is mainly based on Gram Schmidt orthogonalization algorithm to enable the representation of system model using an arbitrary set of functions. This enhances the accuracy of system modeling procedure as it offers a wider selection of functions to represent the system model (Tseng and Powers, 1993). FOS is a modification of orthogonal search that accelerates the process of system modeling through the elimination of some inter-

mediate procedures inside orthogonal search algorithm (Korenberg, 1988). FOS has shown an extensively efficient performance in various applications (Korenberg, 1988, 1989) and accordingly it is employed in this research to enhance the forecasting process of river stream flow based on its reported capabilities in accurate system modeling.

### 1.4   Objectives

The main target of this research is to investigate the potential of utilizing the Fast Orthogonal Search (FOS) method to develop stream-flow forecasting model that achieve consistent and reliable accuracy levels. The proposed method was applied for the Nile River stream-flow at Aswan High Dam AHD for 130 years flow data on a monthly basis during the period between 1870 and 2000. One of the major advantages of the proposed FOS method is its flexibility to manage the data to be applicable for different

training approaches. In this context, three different training approaches have being examined. In addition,



comparative analysis with the previous AI models developed by the authors is carried out and widely discussed. Furthermore, the proposed FOS method is examined for forecasting a complete water year stream-flow on a monthly basis.

The residue of the manuscript is arranged in the following fashion. Section 2 gives the description of the case studies and data-sets. The subsequent section illustrates the methodology part. Results and discussions are given in Section 4. Finally, in Section 5 we conclude the paper and outline our current lines of research.

## 2    Data collection and Pre-Analysis

The current study focuses on developing forecasting model for stream-flow at AHD on Nile River, Egypt Figure (1). The historical natural stream-flow records for 130 years between 1870 and 2000 has been collected from the Nile Water Authority, Ministry of Water Resources and Irrigation, Egypt, which is used in this study. The data for natural stream-flow at AHD experienced two different stages of measurements. The first is during the period between 1870 and 1902 and then for the period between 1903 and 2000. In the first period, the data for the stream-flow has been recorded by the general stage-discharge table. This table was designed from the Aswan downstream gauge. On the other hand, 1903 onward, Egyptian and Sudanese government constructed several dams and hydraulic structures along the Nile River, and then the natural stream-flow records were calculated from the stage-discharge relationship in Aswan Dam fine-tuning the stream-flow records to be free from all the losses from the manmade upstream lakes, water withdraw in Sudan and the influence of the Sennar reservoir.

As it could be depicted from Figure (2) that the collected stream-flow records during the period between 1870 and 2000 are highly random and stochastic in nature. In this context, in the case of developing one of the CMs methods to develop the forecasting model, there is a need to scrutinize the data by investigating the auto-correlation sequence for each month and examine the cross-correlation between





consequent months. However, while utilizing the proposed model FOS, there is no need to examine such correlation analysis or develop a pre-formulation for the data time series.

## 3    Methodology

### 3.1    NARMAX modeling using Fast Orthogonal Search

The NARMAX model is defined as

$$
\begin{aligned}
y(n) = F\big[ & y(n-1), y(n-2), \dots, y(n-n_y), x(n-d), x(n-d-1), \dots, x(n-d \\
& - n_x), e(n-1), e(n-2), \dots e(n-n_e) \big] + e(n)
\end{aligned} \tag{1}
$$

Where $y(n)$, $x(n)$, and $e(n)$ are the system output, input, and noise sequences, respectively; $n_y$, $n_x$ and $n_e$ are the maximum lags for the system output, input, and noise; F[•] is some nonlinear function, and d is a time delay typically set to d = 1. The model is essentially an expansion of past inputs, outputs, and noise terms.

In this research the noise terms $e(n-1), e(n-2), \dots e(n-n_e)$ are neglected to simplify the modeling process. Hence the NARMAX model is reduced to the following form:

$$
\begin{aligned}
y(n) = F\big[ & y(n-1), y(n-2), \dots, y(n-n_y), x(n-1), x(n-2), \dots, x(n-(n_x+1)) \big] \\
& + e(n)
\end{aligned} \tag{2}
$$

The above model terms and their corresponding coefficients are estimated using FOS algorithm. The algorithm uses an arbitrary set of non-orthogonal candidate functions $P_m(n)$ and finds a functional expansion of an input $y(n)$ in order to minimize the mean squared error (MSE) between the input and the functional expansion.

The functional expansion of the input $y(n)$ in terms of the arbitrary candidate functions $P_m(n)$ is given by:



$$y(n) = \sum_{m=0}^{M} a_m \, p_m(n) + e(n) \tag{3}$$

$\{a_m\}_{m=0}^{M}$ are the weights of the functional expansion, and $e(n)$ is the modelling error.

By selecting the candidate functions of non-orthogonal, there is no sole solution for Eq.(2). However, FOS could classify the input with fewer model terms than an orthogonal functional expansion (Korenberg, 1989).

FOS enables the adequate selection of model terms based on the MSE reduction introduced by each term. The selected term is the one that results in the maximum MSE reduction. FOS algorithm can be stopped by 1) setting a limitation on the model maximum number of terms; 2) setting a predefined threshold for the minimum ratio between mean square of the input parameters and the MSE ; 3) examining the MSE reduction introduced by each term and stooping when the maximum MSE reduction of the remaining term is less than the MSE reduction anticipate from a white-noise term (Korenberg, 1989).

NARMAX modeling using FOS is accomplished by selecting candidate functions Pm(n) that are combinations of delayed versions of inputs and outputs. Accordingly, Pm(n) is given by:

$$P_m(n) = \{P_x, P_{xx}, P_y, P_{yy}, P_{xy}\} \tag{4}$$

$P_x$ Represent the set all possible terms of the input with different delays and different orders.

$P_{xx}$ Represent the set all possible combinations of the input terms with different delays and different orders.

$P_y$ Represent the set all possible terms of the output with different delays and different orders.

$P_{yy}$ Represent the set of all possible combinations of the output terms with different delays and different orders

$P_{xy}$ Represent the set of all possible combinations of the input terms and output terms with different delays and different orders

The maximum delay for the inputs and outputs are determined by the training approach.



## 3.2 Model Structure

The first step in developing the forecasting model is to study and analyze the data used to select training (calibration) and testing (validation) data sets. In this context, in this study, the first 10 years of the stream-flow data between 1871 and 1881 showed a more stable trend for annual stream flow as shown in Figure 3. On the other hand, it is essential for the data to be used for training to experience most of the stream flow pattern which was clearly observed in the variation of monthly data over the first 10 years and showed a higher rate of variations for the second 10 years for the period between 1881 and 1891 as shown in Figure 4.

In this research, the prediction of river flow rate is achieved using a polynomial that determines the river flow rate as a function of the month index. Hence the prediction model is given by:

$$\widehat{R}_m{}^N = f^N(m) \tag{5}$$

Where $\widehat{R}_m{}^N$ is the predicted flow rate of the $N^{to}$ year and month ($m = 1,2,.....12$).

## 3.3 Training approach

This approach starts by building a relation between the vectors of actual flow rates of the first year $R_m{}^1$ and month indices $m = 1,2,...12$. Figure 5 (a) illustrates the timeline for the first training approach. The model is improved each year by using the previous years to construct the current year flow rate model $f^N(m)$. The model training of $N^{th}$ year is achieved by feeding the actual vectors of flow rates $R_m{}^n$ ($n = 1,...N-1$) to FOS system identification algorithm to obtain a polynomial relation between month index and river flow rate. Accordingly, FOS creates a model that describes the curve of the annual readings and this model is updated each year.



For the second training approach, a new model is created after a defined number of years and provides a prediction for a fixed number of years. Ten years of data were selected for both training and prediction. The model training for the Nth year is achieved by feeding the vector of flow rates for the previous 10 years (i.e. $R_m{}^n$ $n = N - 10, \dots N - 1$) to FOS system identification algorithm to obtain a polynomial relation between month index and river flow rate for the next 10 years. Therefore FOS builds a new model every ten years independent on the previous models. Figure 5 (b) shows the prediction and training data for 130 years. Note that the first 10 years were used as input hence the prediction is exactly equal to real data.

The third approach is similar to the second approach as it uses the data of the previous years for training to predict the current year river flow rate. The input, in this case, is the flow rate of previous 10 years, and the output is the flow rate of next year. At the first run, the first ten years are used for training and 11th year is predicted. Next, run the 10 years (actual) starting from the second year to 11th year are used for training and 12th year is predicted and so on until year 130. Figure 5 (c) shows the timeline of the third approach for both training and prediction phases.

In order to compare the results of the forecasting obtained with the four algorithms, three different error indicators have been used, calculated over the testing set. Such indicators are the Root Mean Square Error (RMSE) and Relative Error (RE):

$$RMSE = \sqrt{\sum_{i=1}^{N} \frac{(\hat{y}_i - y_i)^2}{N}}, \tag{6}$$

$$NRMSE = \frac{\sqrt{\frac{1}{N}\sum_{t=1}^{n}(\hat{y}_i - y_i)^2}}{y_i} \tag{7}$$

Where $N$ the length of the testing is set and $y_m$ is the average of the time series in the testing set.



## 4    Results and Discussions

The proposed FOS procedure has been adjusted with the data considering the three different training approaches presented in the previous section. Due to the fact that each month has its own pattern as presented in figure (2), the results of the model are structured to be in individual fashion to categorize the results for low, medium and high stream-flow pattern. In addition, the model will be evaluated for each training approach to be able to point out the weakness and strength of each approach in details. In fact, the training procedure is essential step and should be examined in order to understand how the proposed model could achieve accurate forecasting. Actually, having accurate forecasting depend only on the historical stream-flow records, it is vital to assure that the model has been trained with the proper approach. On the other hand, for further assessment, investigating different training approach open the door for examining the potential of the model whether it could achieve accurate forecasting for the long term. According to the previous study at AHD, it was experienced that the accuracy is worsened when the forecasting term goes behind three months ahead.

Furthermore, the performance of the proposed model will be examined for different stream-flow pattern low, medium and high flow. Finally, in order to authenticate whether the forecasted stream-flow harmonize or not with the changes in the stream-flow throughout the testing period, evaluation for the performance index have been carried out.

### 4.1 Training Approach #1

The results of examining FOS model performance utilizing training approach #1 are shown in Figures (6 and 7) for each month. Figure 6 represent the actual and prediction values for 129 years of monthly records.  It can be noticed from Figure 6 that FOS did not have adequate performance while utilizing training approach #1, especially when to examine the peak values where it is clearly there is a significant error to predict the peak value for this scenario. Also, some errors are doubled in most cases and triple times in some odd cases especially that one taken place during the extreme stream-flow events whether





peak or low-stream-flow pattern. The overall performance of this training approach is imprecise where it has (9.7922) for MSE and (-10.5451) for NMSE.

First of all, it should be noticed that twelve months forecasting ahead depending on the previous twelve months is considered to be a very long term stream-flow forecasting. Practically forecasting stream-flow for twelve months ahead are not quite likely and most existing forecasting model support for seasonal time increment. For more exploration of the FOS performance in this scenario, the flow rate has been divided into three groups low, medium, and high. Depending on figures 2 and 3 that shown previously, each four months represent one group where March, April, May and June has the low flow rate over the year. While the high flow rate happened usually at August, September, November, and December. Figure 7 illustrates the performance of FOS in three different scales (low, medium, and high) flow rate. In general, it can be observed that the performance of the FOS model is less efficient with long-term forecasting period.

## 4.2 Training Approach #2

This training approach is different than the previous approach where 10 years are forecasted from the previous 10 years. This type of training is considered as a long term prediction. Figure 8 shows the prediction versus the actual values for 120 years of prediction starting from year 11. By comparing the performance with training approach #1, it can see an improvement on the result where this scenario less MSE (9.0807) and less NMSE (-0.38414). In spite of, the improvement of the result, FOS still encounters difficulty in estimating accurately the peak values. Figure 8 illustrate clearly the underestimation of the peak values. This is due to basically attributed to the increased probability of encountering different stream-flow pattern ranged during a long term of 10 years. Figure 9 shows the actual and prediction values in various scales of stream flow rate. By categorizing the prediction period into three group (high, medium, and low), it can perceive clearly the unsatisfactory performance for FOS in the training approach #2. It can be depicted that the FOS model was trained on the type of stream-flow different from what has been experienced during the period of forecasting. Therefore, the FOS model has not been accurately



updated to mimic this stochastic pattern of the stream-flow. Thus relatively large stream-flow errors of estimation have been reached.

## 4.3 Training Approach #3

This scenario predicting one year ahead based on 10 previous years which is different than the two training approach, therefore, it can be considered as a short-term prediction. The result that has been obtained from this approach, it showed a superior performance than approach #1 and #2. Figure 10 illustrates the prediction versus actual values for 130 years. In spite of, some odd cases have under and over estimation but that not affect the overall performance of FOS prediction (MSE=9.5518 and NMSE=0.050576).

One of the biggest challengings in the prediction of time series is the sudden changes or drop for the data values. Figure 11 shows the actual and prediction data for first 20 years of stream flow. It is noticeable that have a large drop of stream-flow records for each period. FOS shows the ability to overcome that issue in this training approach. It can be depicted that the FOS model was trained on 10 years of stream-flow. Hence, FOS model has been experienced during the period of forecasting. Therefore, the FOS model

has been accurately updated to mimic this stochastic pattern of the stream-flow. Thus, less error of prediction was accomplished.

For more analysis of the FOS performance, Figure 12 shows the forecasting accuracy in different value scales (high, medium, low). The figure illustrates clearly that FOS had a significantly better performance in low and medium stream flow rates than the high flow rates. Finally, we can deduce that the training

approach #3 has better performance than training approach #1 and #2. Therefore, the ten years of input is adequate to predict one year ahead.

In general, it could be noticed from the results and the performance attained when the proposed FOS model with different training approach has been applied as stream-flow forecasting model at AHD that FOS as a technique is pertinent in such hydrological application. The successfulness of FOS to perform

the forecasting mission for stream-flow open the door for more reliable method for stream-flow



forecasting which advance better water resources planning. In addition, the FOS as an algorithm is flexible enough to be tuned for bring applicable for other prediction or forecasting tsks required in water resources and hydrological field such as, evapotranspiration, evaporation, sediment transpose and rainfall. Moreover, FOS algorithm with further enhancement could be not only suitable for time series engineering problems but also for the prediction engineering problem which is considering several input parameters to predict particular output variable.

## 5    Conclusion

In fact, the existing forecasting models utilizing conventional methods or soft computing models do not have standard rules for their implementation nor for their best structure based on the targeted application. In this context, current study attempts to enhance the forecasting accuracy for the river stream flow by proposing a new method namely, Fast Orthogonal Search (FOS). FOS is a robust algorithm for non-linear system modeling .It was employed in this research to enhance the accuracy of forecasting model for river stream flow process. FOS offered a simple and efficient system modeling by using an arbitrary set of functions to represent data patterns.

The proposed FOS model has been examined to accomplish forecasting mission for river stream-flow utilizing 130 years of natural stream-flow historical records on a monthly basis. In order to examine the ability of the proposed FOS technique to be adjustable for different timeline data process, the model has been carried out with three different training approaches. This is due to the fact that in some cases certain training approach could achieve better accuracy than the other based on the nature of data pattern. The results showed that the FOS model could achieve a relatively high level of stream-flow forecasting accuracy for all the stream-flow categories (low, medium and high) while using the third training approach. The third training approach is a sliding window process considering the window size equal to 10 years of training and then switching for forecasting for only one complete year. On the other hand, in this application of AHD, it is of interest to highlight that the proposed FOS model provides a relatively poor level of accuracy for the low category of stream-flow. This might be caused due to the fact that the





natural stream-flow for the low category experienced sudden change and high dynamic fluctuations in its pattern.

Finally, the results showed that there is a potential for proposing the FOS model for long-term forecasting as it could achieve an acceptable level of accuracy for one year ahead forecasting for the natural stream flow. This is particularly significative given that the vast majority of algorithms usually adopted for river flow forecasts could be suited for short-term stream flow forecasting

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



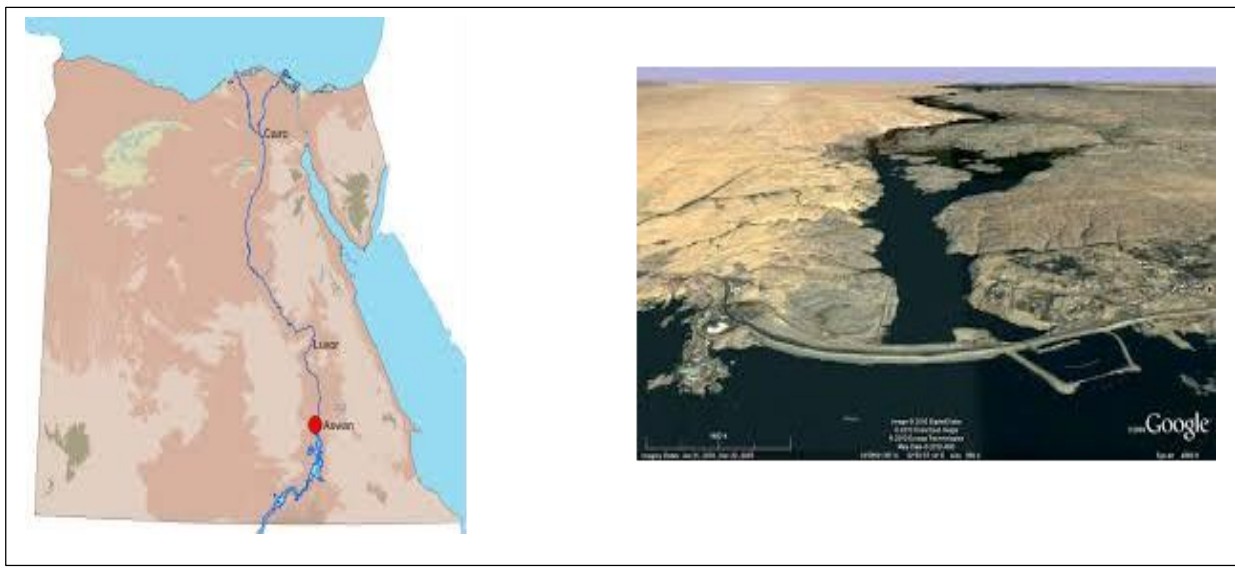

Figure 1: Location of Aswan High Dam, Egypt







Figure 2: Natural Stream flow for Nile River at AHD for 130 years




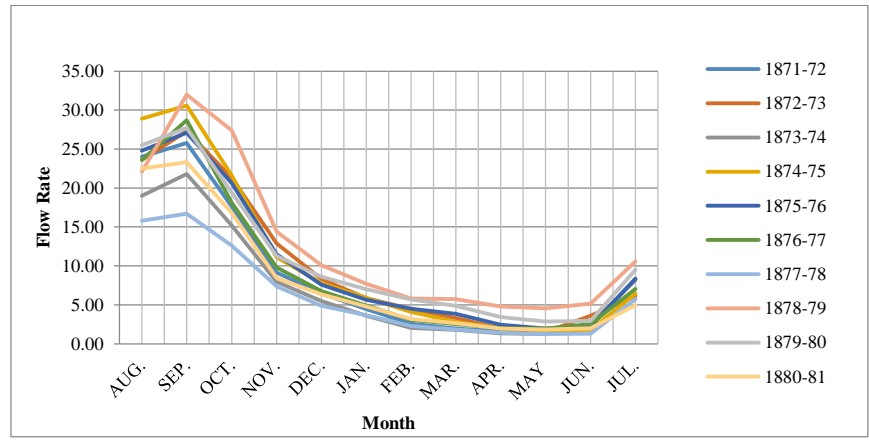

Figure 3 Annual Curves for river flow rate in the perido between 1871 and 1881

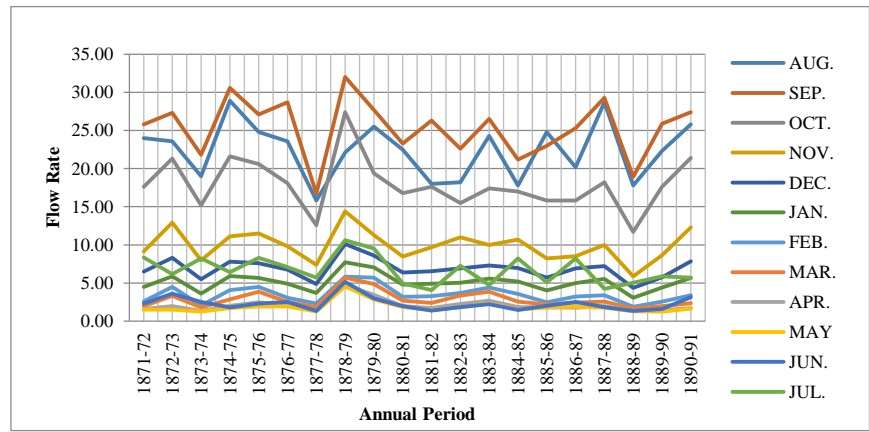

Figure 4 Monthly Curves for river flow rate in the period between 1871 and 1891




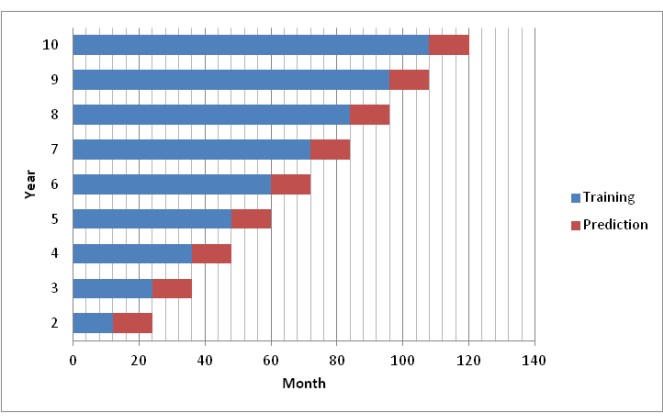

a)  Training Approach #1

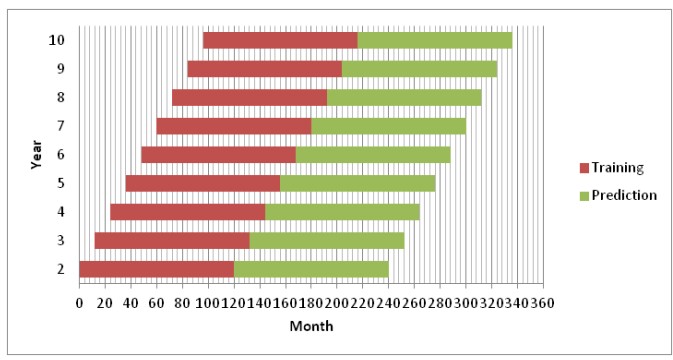

b)  Training Approach #2

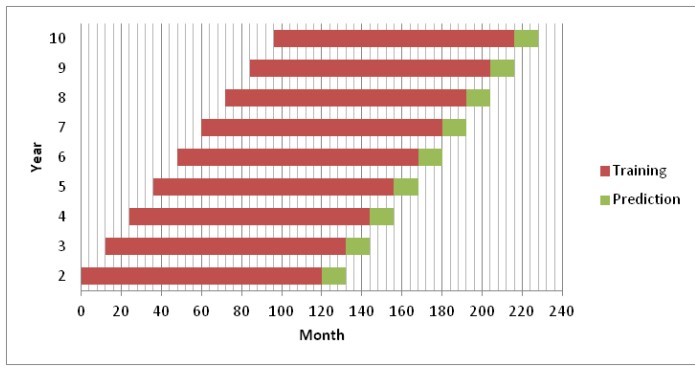

c)  Training Approach #3

Figure 5 Timeline for the different training approach



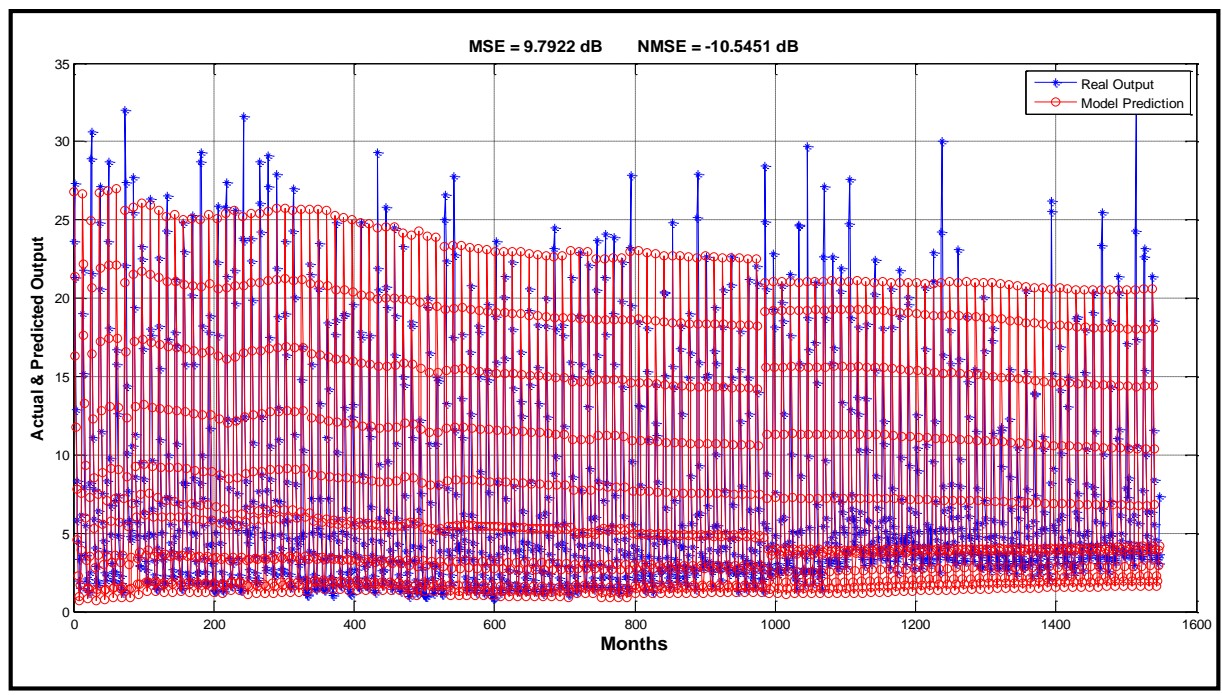

Figure 6 Prediction versus Real output for 129 years for first training approach.

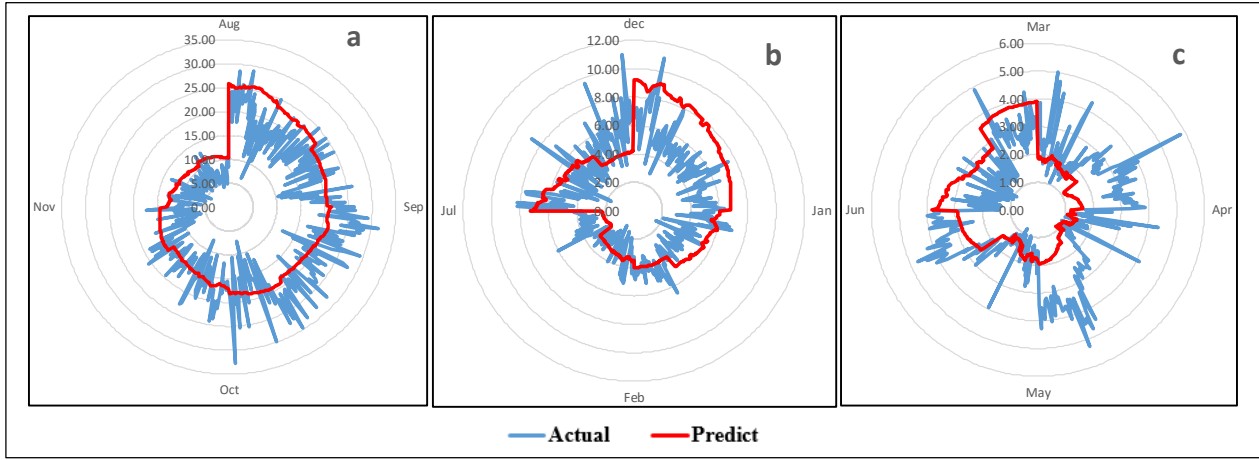

Figure 7 Actual and predicted for a) high b) medium c) low flow rate for training approach #1.





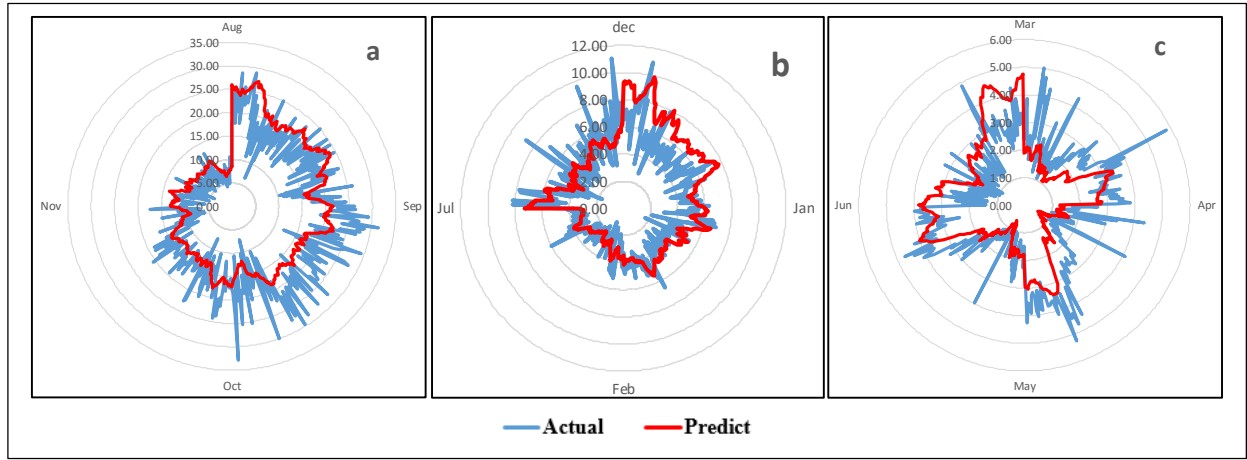

Figure 8 Prediction versus Real output for first 120 years (from year 11 to year 130)

Figure 9 Actual and predicted for a) high b) medium c) low flow rate for training approach #2.




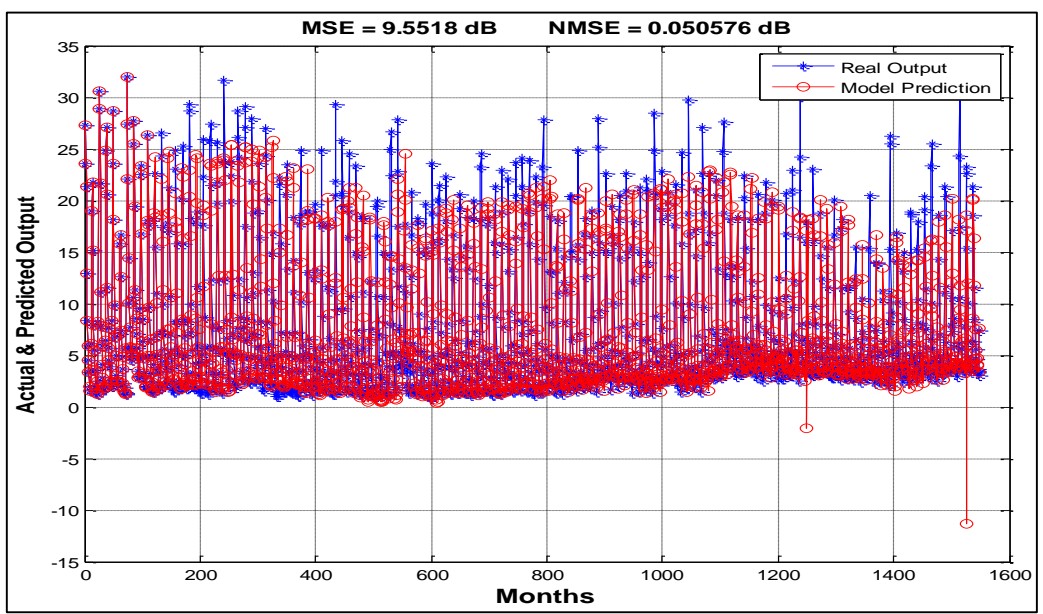

Figure 10 Prediction versus Real O/P for first 130 year

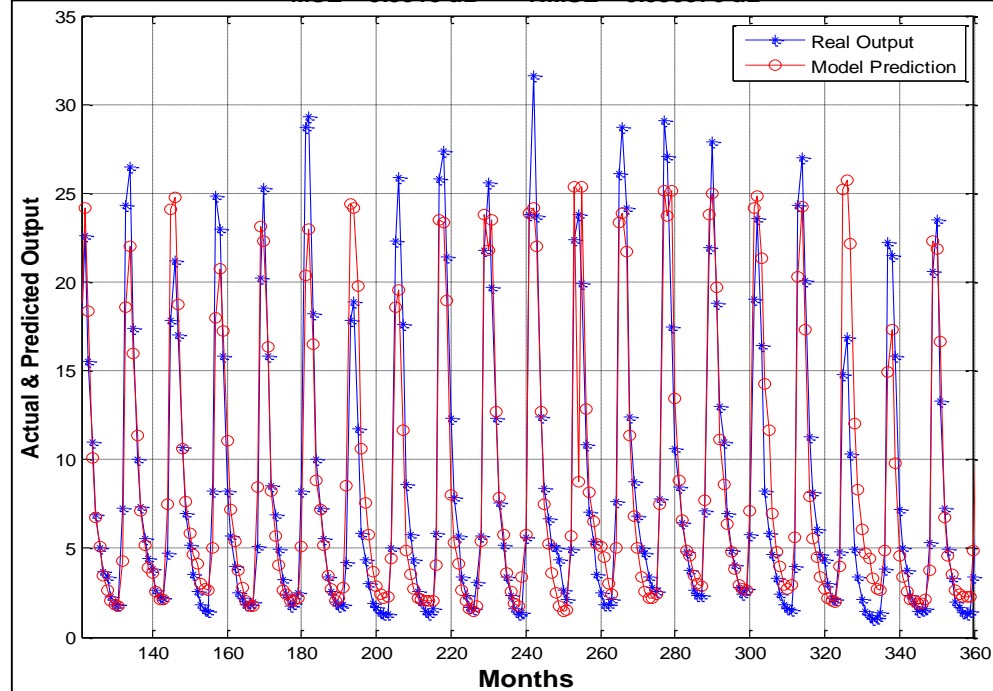

Figure 11 Prediction versus Real O/P for first 20 years (from year 11to year 30)





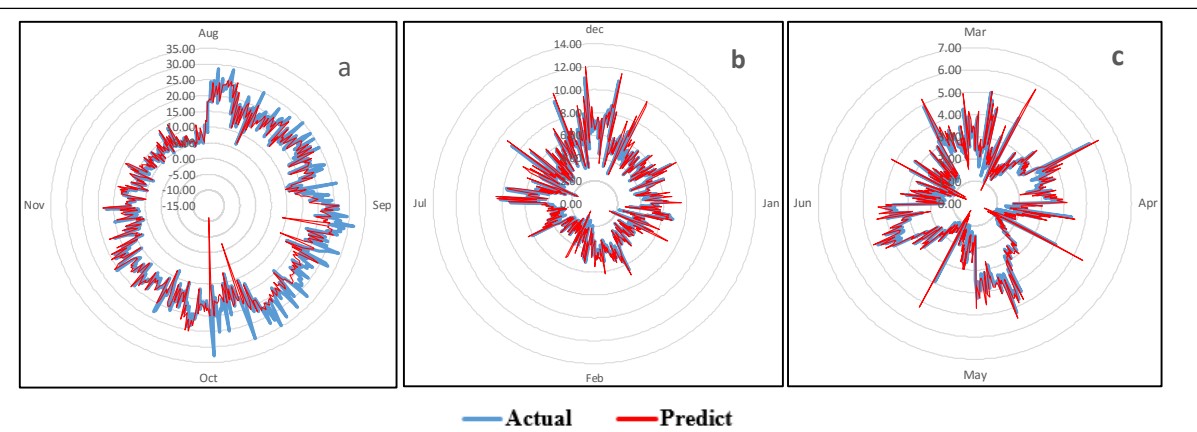

Figure12 Actual and predicted for a) high b) medium c) low flow rate for training approach #3.