# Peer review of "Acclimatizing Fast Orthogonal Search (FOS) Model for River Stream-flow Forecasting"

_Hydrology and Earth System Sciences, 2016_

## Referee Comment (RC1) · Anonymous Referee #1 · 24 Aug 2016

The manuscript presents a new method namely Fast Orthogonal Search (FOS) for stream-flow forecasting, which is interesting. The model is tested using stream-flow data at Aswan High Dam located in Egypt. The subject is within the scope of the journal. Overall, I think the paper is well written and the authors address an important topic in hydrology (stream-flow forecasts) that is of keen interest of the Hydrology community Based solely on the paper results, I am fully convinced that the proposed FOS model has much advantage over the classical AR and ARMA (including periodic AR) even the Artificial Neural Network model classes of stream-flow forecast models. However, some results are NOT addressed and discussed adequately. Moreover, the apparent relative medium forecasting skill of the proposed model needs to be discussed. The manuscript, in its present form has the potential for publication in HESS with adequate revisions to the following points which should be undertaken in order to justify recommendation for publication. • For readers to quickly catch the contribution in this work, it would be better to highlight major difficulties and challenges, and your original achievements to overcome them, in a clearer way in abstract. • Many assumptions are stated in various sections. More justifications should be provided on these assumptions. Evaluation on how they will affect the results should be made. • The key FOS parameters are not mentioned. The rationale on the choice of the particular set of parameters should be explained. Have the authors experimented with other sets of values? What are the sensitivities of these parameters on the results? • It is mentioned in p.4 line 17-19 that "Even though these AI models demonstrate to be proficient, the convergence of the model during the training (calibration) experienced a slow procedure which means that the model falls in the sub-optimal search procedure.". Some justifications should be furnished on this. • There is missing information about the major feature of the Nile basin and proper statistical analysis for the data. It is vital for the readers to get complete information about the basin and also brief statistical analysis for the raw data. • The authors presents the three different training approaches for the model and shows graphically its time-line procedure, however there is absence of the major mode structure for the model, even the authors describe the model structure satisfactorily, it would be better to show a block diagram for the model structure. • It would be of importance for the readers to see more performance indicators for the model evaluation to be presented. In addition, as long as the authors are presenting the results on monthly basis, it would be better to do so for each month. • Moreover, the manuscript could be substantially improved by relying and citing more on recent literatures about case studies of application of various types of soft computing technique in discharge prediction such as the followings:

- Cheng, C.T., Wu, X.Y. and Chau, K.W., "Multiple criteria rainfall-runoff model calibration using a parallel genetic algorithm in a cluster of computer," Hydrological Sciences Journal, Vol. 50, No. 6, 2005, pp. 1069-1087.

- Lin, J.Y., Cheng, C.T. and Chau, K.W., "Using support vector machines for long-term

discharge prediction," Hydrological Sciences Journal, Vol. 51, No. 4, 2006, pp. 599-612.

- Wang, W.C., Chau, K.W., Cheng, C.T. and Qiu, L., "A comparison of performance of several artificial intelligence methods for forecasting monthly discharge time series," Journal of Hydrology, Vol. 374, No. 3-4, 2009, pp 294-306.

- Wu, C.L., Chau, K.W. and Li, Y.S., "Predicting monthly streamflow using data-driven models coupled with data-preprocessing techniques," Water Resources Research, 45, W08432, doi:10.1029/2007WR006737, 2009.

- Cheng, C.T., Ou, C.P. and Chau, K.W., "Combining a fuzzy optimal model with a genetic algorithm to solve multiobjective rainfall-runoff model calibration," Journal of Hydrology, Vol. 268, No. 1-4, 2002, pp. 72-86.

- Chau, K.W., "Particle swarm optimization training algorithm for ANNs in stage prediction of Shing Mun River," Journal of Hydrology, Vol. 329, No. 3-4, 2006, pp. 363-367.

• Complete results for all the performance indicators should be presented in the discussion section.

• In the conclusion section, the limitations of this study, suggested improvements of this work and future directions should be highlighted.

---

## Short Comment (SC1) · 17 Sep 2016

Acclimatizing Fast Orthogonal Search (FOS) Model for River Stream-flow Forecasting
Abdalla Osman, Mohammed Falah Allawi, Haitham Abdulmohsin Afan , Aboelmagd
Noureldin, Ahmed El-shafie

We would like to thank the referee for his objective and thorough review of our paper.
We have addressed all the referee's comments in the following point-by-point response.
All changes made to accommodate the referee's comments are underlined in the re-
vised manuscript.

Reviewer #1

- The manuscript presents a new method namely Fast Orthogonal Search (FOS) for

inflow forecasting, which is interesting. The model is tested using inflow data from the Aswan High Dam located in Egypt. The subject addressed is within the scope of the journal. Overall, I think the paper is well written and the authors address an important topic in hydrology (inflow forecasts) that is of keen interest of the Hydrology community Based solely on the paper results, I am not fully convinced that the proposed FOS model has much advantage (if any) over the classical AR and ARMA (including periodic AR) even the Artificial Neural Network model classes of inflow forecast models. Some results are NOT addressed and discussed adequately. Moreover, the apparent relative medium forecasting skill of the proposed model needs to be discussed. However the manuscript, in its present form has the potential for publication in HESS with adequate revisions to the following points which should be undertaken in order to justify recommendation for publication.

Reply

The author thanks the reviewer for his comments. The authors address all of his comments one-by-one hereafter and modify the manuscript

-For readers to quickly catch the contribution in this work, it would be better to highlight major difficulties and challenges, and your original achievements to overcome them, in a clearer way in abstract.

Reply

Owing to the referee feedback, challenges and difficulties about the flow forecasting and also the original achievements have been reported more clearer way in the introduction section.

- Many assumptions are stated in various sections. More justifications should be provided on these assumptions. Evaluation on how they will affect the results should be made.

Reply

It is true that there are some assumptions in our research. Hereafter, we will try to highlight the major ones. Assume the training approach. The findings of the cross-correlation analysis for the monthly natural inflow pattern for consequences years shows that the cross-correlation is relatively poor if go more than one year behind the one under study to be forecasted for most of the months. Based on that observation, theoretically, for the forecasting model has been developed based on the previous year inflow pattern for all training approaches used as training period and the followed forecasting period. Assumed performance indicators Actually, in developing such forecasting model utilizing time series concept, the model could perform well during the training period and might provide higher level of error when evaluating during either validation or testing period. In this context, in this study the authors used these performance indices to make sure of that the proposed model could provide consistent level of accuracy during all periods. The advantages of utilizing these two statistical indices as a performance indicator of the proposed model are as follow:- Using the maximum error is to make sure that the highest error while evaluating the performance is within the acceptable error for such forecasting model. While utilizing the Root Mean Square error is to ensure that the summation of the error distribution within the validation period is not high. Consequently, using both indices is guaranteed consistent level of errors which is providing a great potential for having same level error while examining the model for unseen data in the testing period.

- The key FOS parameters are not mentioned. The rationale on the choice of the particular set of parameters should be explained. Have the authors experimented with other sets of values? What are the sensitivities of these parameters on the results?

Reply

In fact, there is no formal and/or mathematical method for determining the appropriate "optimal set" set of the key parameters of FOS which are Model Order, Maximum Delay, Mean Square Error and Mean Square Error Reduction). Accordingly, the authors decide to perform this task utilizing trial and error method. The authors experimented

several sets and examined each experiment but we report only the best trial. However, the authors reported some observations about the proposed model performance and sensitivity analysis under different set of key parameters in the revised version of the manuscript. Hereafter the details of those four parameters: "There are four main parameters governing the NARMAX model formed by FOS. The First two parameters are the model order and the maximum delay of the NRMAX model. The maximum model order in this research was initially set to 5 and the maximum model order obtained was 3. The maximum delay was set to 12 and was decided based on the data feeding process. The maximum delay of the obtained models did not exceed 6. The third parameter is the minimum MSE obtained and this was set to ãĂŰ10ãĂŮ^(-6) to ensure best fitting of data. The fourth parameter is the MSE reduction introduced by each term. This parameter value is internally determined by FOS."

- It is mentioned in p.4 line 17-19 that "Even though these AI models demonstrate to be proficient, the convergence of the model during the training (calibration) experienced a slow procedure which means that the model falls in the sub-optimal search procedure.". Some justifications should be furnished on this.

Reply

The authors fully agreed with the referee in this point that the statement is not fully understandable. In this context, the authors add more clarification in this position of the manuscript. In most of Artificial Neural Network (ANN) models development the back-propagation algorithm is used for optimizing the ANN key parameters. The back-propagation algorithm experienced several drawbacks such as, local optima, slowness. In fact, there are many advanced methods offered by researchers to overcome partially these drawbacks especially the local optima such as Particle Swarm Optimization (PSO) and Genetic Algorithm (GA). However, utilizing those proposed optimization algorithm to treat the drawback of the back-propagation experienced another challenges such as over-fitting problem for the whole model performance.

- There is missing information about the major feature of the Nile basin and proper statistical analysis for the data. It is vital for the readers to get complete information about the basin and also brief statistical analysis for the raw data.

Reply

Owing to the reviewer feedback, the authors add comprehensive description for the Nile River Basin has been reported in the case study section. In addition, statistical analysis for the natural inflow pattern for 130 years at AHD has been carried out.

- The authors presents the three different training approaches for the model and shows graphically its time-line procedure, however there is absence of the major mode structure for the model, even the authors describe the model structure satisfactorily, it would be better to show a block diagram for the model structure.

Reply

Block diagram for the model structure has been added (Figure )

- It would be of importance for the readers to see more performance indicators for the model evaluation to be presented. In addition, as long as the authors are presenting the results on monthly basis, it would be better to do so for each month.

Reply

Owing to the reviewer feedback, the authors add one more table " table " to show the complete performance for the proposed FOS model showing the performance indicators (four performance indicators" for each month.

- Moreover, the manuscript could be substantially improved by relying and citing more on recent literatures about case studies of application of various types of soft computing technique in discharge prediction such as the followings: - Cheng, C.T., Wu, X.Y. and Chau, K.W., "Multiple criteria rainfall-runoff model calibration using a parallel genetic algorithm in a cluster of computer," Hydrological Sciences Journal, Vol. 50, No. 6,

2005, pp. 1069-1087. - Lin, J.Y., Cheng, C.T. and Chau, K.W., "Using support vector machines for long-term discharge prediction," Hydrological Sciences Journal, Vol. 51, No. 4, 2006, pp. 599-612. - Wang, W.C., Chau, K.W., Cheng, C.T. and Qiu, L., "A comparison of performance of several artificial intelligence methods for forecasting monthly discharge time series," Journal of Hydrology, Vol. 374, No. 3-4, 2009, pp 294-306. - Wu, C.L., Chau, K.W. and Li, Y.S., "Predicting monthly streamflow using data-driven models coupled with data-preprocessing techniques," Water Resources Research, 45, W08432, doi:10.1029/2007WR006737, 2009. - Cheng, C.T., Ou, C.P. and Chau, K.W., "Combining a fuzzy optimal model with a genetic algorithm to solve multiobjective rainfall-runoff model calibration," Journal of Hydrology, Vol. 268, No. 1-4, 2002, pp. 72-86. - Chau, K.W., "Particle swarm optimization training algorithm for ANNs in stage prediction of Shing Mun River," Journal of Hydrology, Vol. 329, No. 3-4, 2006, pp. 363-367.

Reply

All the above references have been reviewed and included in the revised manuscript.

- Complete results for all the performance indicators should be presented in the discussion section.

Reply

The authors improve the results and discussion section adding more details discussion on the model performance.

- In the conclusion section, the limitations of this study, suggested improvements of this work and future directions should be highlighted.

Reply

The conclusion section has been improved and includes the limitations of this study, suggested improvements of this work and future directions.

---

## Referee Comment (RC2) · Anonymous Referee #2 · 7 Nov 2016

The paper presents an application of one of the data-driven approaches to monthly flow forecasting of the River Nile flow at Aswan High Dam. The authors state that the objective of their paper is to "investigate the potential of utilizing the Fast Orthogonal Search (FOS) method to develop streamflow forecasting model that achieve consistent and reliable accuracy levels".

The title of the paper and its objective stated by the authors suggest that using FOS is a requirement for developing a consistent and reliable forecasting model, whilst in truth it is just one of many possible techniques used in system identification and for training neural network-based models. For example, the authors could use Adaptive Orthogonal Search (Billings and Wei, 2008) for the same purpose. The paper is not clearly written and rather confusing. The authors formulate the forecasting problem as the Nonlinear Auto-Regressive Moving Average with eXogeneous inputs NARMAX

model, but only one variable, flow rate is used. Therefore, the authors probably use a Nonlinear Auto-Regressive NLAR model. However, the reader can only guess which model is used because it is not presented in any detail and the authors themselves call it the FOS model.

The authors test three different training approaches for the identification of the polynomial relationship between past and future monthly flow rates using FOS system identification tools. There is no information on which computer package is used. The application of FOS is new in flow forecasting, at least to my knowledge, although it is a well-known technique in engineering applications. The authors present one application to the monthly flow forecasting of the River Nile, but they do not provide any new insight into the subject. In other words, it is not clear if this approach could be useful for other rivers and what we can learn from using it.

Apart from the very poor language that requires serious editing, the paper could serve as a caricature of a scientific paper. I do not think the authors read what they wrote. There is a number of repetitions, the authors give lists of references that are not relevant, statements are wrong or meaningless (see specific comments). I suggest the authors refer to the paper of Billings and Wei (2008) to improve their paper-writing skills and submit a corrected version of the paper as a technical note to some other journal. In my opinion, the paper is not suitable for publication in HESS.

Reference

Billings, S.A., Wei, H., 2008. An adaptive orthogonal search algorithm for model subset se-lection and nonlinear system identification. Int. J. Control. 81, 714-724.

Specific comments:

The title is not precise: I guess the authors meant "river flow forecasting". I would suggest change "stream-flow" into "river flow" in the rest of the paper.

Page 1, lines 19-20: "In this paper, a novel model namely; Fast Orthogonal Search

(FOS) model is proposed to develop river stream-flow forecasting." This sentence states the FOS is novel which is not true. It was first published in 1989 by Korenberg (referred to by the authors) and it is not a model but an algorithm for system identification.

Page 2, line 9-10 repeats line 8-9.

Page 3, lines 10-20 and page 4 lines 1-10 are only two examples of very poor writing style mentioned in the general comments.

Page 4, equation (5): the f(m) function is not specified and therefore the model structure is not given.

---

## Author Comment (AC2) · 23 Nov 2016

**Acclimatizing Fast Orthogonal Search (FOS) Model for River Stream-flow Forecasting**

**We would like to thank the referee for his objective and thorough review of our paper. We have addressed all the referee's comments in the following point-by-point response. All changes made to accommodate the referee's comments are underlined in the revised manuscript.**
* * *
**Reviewer #2**

The paper presents an application of one of the data-driven approaches to monthly flow forecasting of the River Nile flow at Aswan High Dam. The authors state that the objective of their paper is to "investigate the potential of utilizing the Fast Orthogonal Search (FOS) method to develop stream-flow forecasting model that achieve consistent and reliable accuracy levels".

The title of the paper and its objective stated by the authors suggest that using FOS is a requirement for developing a consistent and reliable forecasting model, whilst in truth it is just one of many possible techniques used in system identification and for training neural network-based models. For example, the authors could use Adaptive Orthogonal Search (Billings and Wei, 2008) for the same purpose.

**Reply**

**The objective of the paper is to apply non-linear system identification to the problem of river stream-flow forecasting and compare its performance with Artificial Intelligence (AI) forecasting techniques. For this purpose, any powerful non-linear system identification technique can provide a proof of concept. However, the selection of FOS was motivated by its capabilities and its performance that was previously reported in (Korenberg, M. J. 1988, Osman et al (2010), Osman et al (2009)). FOS has been developed, validated and examined by the authors in variety of other applications and showed superior performane over other competitive methods. Other methods such as the "The Adaptive Orthogonal Search (Billings and Wei, 2008)" can be compared later to FOS to asses the performance of different non-linear system identification techniques which is not the main scope of this paper. For example, the main difference between FOS and AOS is the model terms selection and the model length determination criteria. We will consider carrying out this comparison in another publication to assess the**

**performance of orthogonal search methods and to evaluate FOS against other non-linear system identification techniques.**

Osman A., **Noureldin A.**, El-Shafie A. and McGaughey D.: "Fast Orthogonal Search Approach for Distance Protection of Transmission Lines" Electric Power Systems Research, Elsevier, **V80** (2), pp: 215–221, Feb 2010.

*Osman A.*, **Noureldin A.**, El-Sheimy N., Theriault J. and Campbell S. "Improved Target Detection and Bearing Estimation Using Fast Orthogonal Search for Real-Time Spectral Analysis" Measurement Science and Technology, IoP, **V20** (6), June 2009 (14pp).

The paper is not clearly written and rather confusing. The authors formulate the forecasting problem as the Nonlinear Auto-Regressive Moving Average with eXogeneous inputs NARMAX model, but only one variable, flow rate is used. Therefore, the authors probably use a Nonlinear Auto-Regressive NLAR model. However, the reader can only guess which model is used because it is not presented in any detail and the authors themselves call it the FOS model.

**Reply**

**We thank the reviewer for this important comment and for pointing out this issue that may confuse the reader. In addition to proof reading to improve its quality, the paper has been better organized so that the above confusion is resolved. A paragraph is added and is underlined in the revised manuscript to clearly explain that, in this particular application, the FOS based model was reduced from NARMAX to a nonlinear autoregressive (NLAR) model. For the reviewer's convenience, the added part is also included below.**

The authors test three different training approaches for the identification of the polynomial relationship between past and future monthly flow rates using FOS system identification tools. There is no information on which computer package is used.

**Reply**

**The authors used MatLab to implement the FOS algorithm on a Core i7 (3.4 GHz) processor utilizing a 16GB RAM. This information is added in the revised manuscript.**

The application of FOS is new in flow forecasting, at least to my knowledge, although it is a well-known technique in engineering applications. The authors present one application to the monthly flow forecasting of the River Nile, but they do not provide any new insight into the subject. In other words, it is not clear if this approach could be useful for other rivers and what we can learn from using it.

**Reply**

**It is true that the utilization of orthogonal search techniques including FOS for stream flow forecasting is new and has not been applied before. The scope of the manuscript is to show the potential of FOS as forecaster for the river stream-flow. The same approach can be applied to other rivers. We have added two sentences in the conclusion section to explain the potential of using FOS in stream flow forecasting of other rivers. The added sentences is underlined in the revised manuscript.**

Apart from the very poor language that requires serious editing, the paper could serve as a caricature of a scientific paper. I do not think the authors read what they wrote. There is a number of repetitions, the authors give lists of references that are not relevant, statements are wrong or meaningless (see specific comments). I suggest the authors refer to the paper of Billings and Wei (2008) to improve their paper-writing skills and submit a corrected version of the paper as a technical note to some other journal.

**Reply**

**We thanks the reviewer for his observation and for commenting on the quality of the language of the paper. We revised the paper carefully to address his comment. It has gone through significant proof read and reorganization in order to improve both the structure and the language of the paper. Moreover, the list of references has been revised as per the reviewer comment. It has been modified to be more comprehensive and reprehensive to the citation made on the paper.**

**Specific comments:**

The title is not precise: I guess the authors meant "river flow forecasting". I would suggest change "stream-flow" into "river flow" in the rest of the paper.

**Reply**

**We thank the reviewer for his comment. However, the authors believe that the term stream-flow represents the flow of water in streams, rivers, and other channels, as defined the global hydrological cycle.**

Page 1, lines 19-20: "In this paper, a novel model namely; Fast Orthogonal Search (FOS) model is proposed to develop river stream-flow forecasting." This sentence states the FOS is novel which is not true. It was first published in 1989 by Korenberg (referred to by the authors) and it is not a model but an algorithm for system identification.

**Reply**

**The authors agree with the reviewer in his comment. We revised the paper to show a novel forecasting approach based on Fast Orthogonal Search (FOS).**

Page 2, line 9-10 repeats line 8-9.

Page 3, lines 10-20 and page 4 lines 1-10 are only two examples of very poor writing style mentioned in the general comments.

Page 4, equation (5): the f(m) function is not specified and therefore the model structure is not given.

**Reply**

**The manuscript has been thoroughly reviewed and all the efforts have been made to make it free from errors. We have also made all necessary efforts to improve the structure and the organization of the paper and avoid any repetition.**